# Sleep does not influence schema-facilitated motor memory consolidation

**Serena Reverberi** [1,2], **Nina Dolfen**[1,2], **Anke Van Roy**[3], **Genevieve Albouy**[1,2,3]*, **Bradley R. King**[3]

**1** Department of Movement Sciences, Motor Control and Neural Plasticity Research Group, KU Leuven, Leuven, Belgium, **2** LBI—KU Leuven Brain Institute, KU Leuven, Leuven, Belgium, **3** Department of Health and Kinesiology, College of Health, University of Utah, Salt Lake City, UT, United States of America

* genevieve.albouy@kuleuven.be

## Abstract

### Study objectives

Novel information is rapidly learned when it is compatible with previous knowledge. This "schema" effect, initially described for declarative memories, was recently extended to the motor memory domain. Importantly, this beneficial effect was only observed 24 hours–but not immediately–following motor schema acquisition. Given the established role of sleep in memory consolidation, we hypothesized that sleep following the initial learning of a schema is necessary for the subsequent rapid integration of novel motor information.

### Methods

Two experiments were conducted to investigate the effect of diurnal and nocturnal sleep on schema-mediated motor sequence memory consolidation. In Experiment 1, participants first learned an 8-element motor sequence through repeated practice (Session 1). They were then afforded a 90-minute nap opportunity (N = 25) or remained awake (N = 25) before learning a second motor sequence (Session 2) which was highly compatible with that learned prior to the sleep/wake interval. Experiment 2 was similar; however, Sessions 1 and 2 were separated by a 12-hour interval that included nocturnal sleep (N = 28) or only wakefulness (N = 29).

### Results

For both experiments, we found no group differences in motor sequence performance (reaction time and accuracy) following the sleep/wake interval. Furthermore, in Experiment 1, we found no correlation between sleep features (non-REM sleep duration, spindle and slow wave activity) and post-sleep behavioral performance.

### Conclusions

The results of this research suggest that integration of novel motor information into a cognitive-motor schema does not specifically benefit from post-learning sleep.

**Data Availability Statement:** Source data corresponding to figures and tables in this manuscript, as well as raw data, are publicly available on Zenodo at the following link: https://doi.org/10.5281/zenodo.7347842.

**Funding:** Funding from the FWO Research
Foundation Flanders (G0B1419N) supported this
work. GA also received additional support from the
FWO Research Foundation (G099516N,
G0D7918N, G0B1419N, 1524218N) and internal
funds from KU Leuven. SR was supported by a
fellowship from the FWO Research Foundation
(11C6221N).The funders had no role in study
design, data collection and analysis, decision to
publish, or preparation of the manuscript.

**Competing interests:** The authors have declared
that no competing interests exist.

## Introduction

When new information is learned, it can be rapidly integrated into memory if it is compatible with preexisting knowledge. This is referred to as the memory "schema" effect. A schema can be conceptualized as an associative knowledge structure that represents a general concept abstracted from previous experiences, and it is believed to be stored in the neocortex [1]. Evidence that cognitive schema can accelerate the learning of novel information was first provided more than a decade ago in rats [2, 3] and has since been demonstrated in humans using a variety of declarative memory tasks, including associative [4], sentence [5] and melody [6] learning. These previous studies revealed that new melodies are easily learned when they conform to a known tonal schema [6]; new facts are easily acquired by students when they conform to their educational background [5]; and new object-scene associations are easily remembered when consistent with real-world knowledge (e.g. an umbrella and a rainy scene) [4].

Recently, the schema effect was extended to the motor memory domain [7]. The results of this large behavioral study suggested that during motor learning a cognitive motor schema develops which encompasses the binding between movements and their ordinal position. Importantly, performance was significantly enhanced when new movements were embedded in a sequence whose ordinal structure was compatible, as compared to incompatible, with that of the previously learned motor sequence. These findings indicated that an acquired cognitive-motor schema consisting of the ordinal structure of the originally-learned sequence facilitated the learning of a novel sequence of movements.

Interestingly, this schema effect was only observed if the initial memory trace had enough time to consolidate. Specifically, the schema-mediated integration of new motor memories was only observed 24 hours, but not immediately, after initial learning. Given the established role of sleep in memory consolidation [8–10], it could be speculated that sleep is necessary for the consolidation of cognitive motor schema in which new information can subsequently be integrated. Such a possibility would be consistent with previous research in the declarative memory domain. Sleep, and specific plasticity-related sleep features such as spindles and slow waves that are present in non-rapid-eye-movement (NREM) sleep, are thought to support declarative schema formation and subsequent integration of novel information [[1, 11–15] although see [16]]. While our previous results hinted that sleep may be crucial for the integration of novel motor information into a previously-acquired schema [7], the design employed could not differentiate the specific effects of sleep from the influence of time.

In the current research, we conducted two separate experiments investigating whether diurnal (Experiment 1) or nocturnal (Experiment 2) sleep following the acquisition of a motor memory schema is necessary for the subsequent fast integration of novel, schema-compatible motor information. We hypothesized that participants who were allowed a post-learning sleep opportunity would display enhanced integration (i.e., decreased response time) compared with participants who remained awake. Furthermore, we hypothesized that time spent in NREM sleep, as well as spindle and slow wave activity during the post-learning sleep epoch, would positively correlate with measures of memory integration and memory retention.

## Methods

Experiment 1 examined the effects of diurnal sleep on schema-mediated integration processes and thus consisted of a Nap/No-Nap experimental design. Data collection and analysis plans were pre-registered via Open Science Framework and can be accessed at https://osf.io/7gxm3. Data collection and the primary statistical analyses did not deviate from the original registered plan. Any additional analyses not included in the pre-registration are labelled in this text as exploratory. Experiment 2 examined the effects of nocturnal sleep on integration processes

and consisted of an AM-PM/PM-AM design. Although not formally pre-registered, the participant inclusion/exclusion criteria, motor tasks and data analyses were identical to the pre-registered Experiment 1.

Source data corresponding to figures and tables in this manuscript, as well as raw data, are publicly available on Zenodo at the following link: https://doi.org/10.5281/zenodo.7347842.

## Participants

Young (age range 18–30 years old) healthy volunteers of all genders were recruited from KU Leuven and the surrounding communities. Inclusion criteria were: 1) right-handed, as assessed by the Edinburgh Handedness Inventory [17], 2) no prior extensive training with a musical instrument requiring dexterous finger movements (e.g., piano, guitar) or as a professional typist, 3) free of medical, neurological, psychological, or psychiatric conditions, including depression and anxiety as assessed by the Beck's Depression and Anxiety Inventories [18, 19], 4) no indications of abnormal sleep, as assessed by the Pittsburgh Sleep Quality Index (PSQI [20]; 5), not considered extreme morning or evening types, as quantified with the Horne & Ostberg chronotype questionnaire [21], 6) free of psychoactive and sleep-influencing medications, 7) non-smokers, and 8) no trans-meridian trips or work night shifts in the three months prior to participation. One hundred and sixteen individuals met these inclusion criteria (across the two experiments) and thus initiated participation in an experimental session. All participants gave written informed consent before the start of the study and received compensation for their participation.

As the specific effects of sleep on the integration of novel motor information into memory has not been previously investigated, a literature-based a priori estimation of the hypothesized effect size is not possible. Nonetheless, given the identical task and an analogous experimental design, sample sizes were determined on the basis of the effect size measured in our previous work investigating the influence of compatible vs. incompatible ordinal structure on motor memory integration [7]. The primary effect of interest in this earlier work was the group main effect in the response time (RT) for novel transitions in the session 2 training phase, as assessed with a 2 (compatible vs. incompatible) x 20 (practice blocks) ANOVA. This group main effect was significant ($F_{(1,36)} = 5.29$, $p = 0.027$, $\eta p2 = 0.128$ / Cohen's F = 0.383). Corresponding power analysis (Effect size f = 0.383, tails = 2; alpha = 0.05, Power = 0.80, correlation among repeated measures = 0.72) via the software G*Power [22] resulted in an estimated minimum of 21 subjects per group. In Experiment 1, we aimed to include 25 subjects per group to be slightly more conservative. To achieve this desired sample, 55 participants started the protocol. Five individuals were excluded from all data analyses due to not completing the experiment (n = 4) or failure to comply to experimental instructions (n = 1), resulting in a final sample size of N = 50 (34 females). Note that two subjects were excluded from analyses of Session 2 test runs due to a keyboard malfunction during one or more blocks of this run.

In Experiment 2, the estimated sample size was increased by 20% to account for higher between-subject variance as task completion was not completed in a controlled laboratory environment (as opposed to our previous work) but at the participants' own home via the web-based data acquisition platform Psytoolkit [23, 24]. Of the 61 participants who started Experiment 2, 1 was excluded from data analysis for failure to comply to experimental instructions, resulting in the planned sample size of 60 participants. Following inspection of the data, a further 3 participants (1 belonging to the AM-PM, 2 to the PM-AM group) were excluded from analyses as they failed to appropriately perform the motor task (i.e., statistical outliers (> 3 SD from group mean) on sequence accuracy during Session 1), resulting in a final sample size for analyses of N = 57 (40 females).

## General experimental procedures

In both experiments, participants completed two sessions of motor tasks (described below), separated by a delay of either approximately 4 (Experiment 1; Fig 1A) or 12 hours (Experiment 2; Fig 1B). During the first experimental session (Session 1), participants learned a specific motor sequence (Sequence 1). In the second experimental session (Session 2), integration of new movements into the motor sequence knowledge developed in Session 1 was assessed (see details below). Importantly, the effect of sleep on integration processes was assessed with the comparison between a sleep group (diurnal and nocturnal in Experiments 1 and 2, respectively) and a matched wake control group (see below for details). At the start of each session, the psychomotor vigilance task (PVT [25]) and Stanford Sleepiness Scale (SSS [26]) were administered to provide objective and subjective assessments of vigilance, respectively. Additionally, the beginning and end of Sessions 1 and 2, respectively, included runs of a pseudorandom Serial Reaction Time Task (SRTT) variant (see subsequent section for details), allowing the assessment of general motor execution independent of sequence learning per se.

All participants followed a constant sleep/wake schedule (according to their own rhythm) for 4 days (3 nights) prior to the first experimental session. Compliance was assessed with self-reported sleep diaries, as well as wrist actigraphy recordings (ActiGraph, Pensacola, USA; Experiment 1 only). Sleep quality and quantity for the night preceding the experimental sessions were assessed with the St. Mary's sleep questionnaire [27].

Participants were instructed to avoid alcohol, caffeine, daytime sleeping/napping, and heavy exercise during the 12h preceding each experimental session, and to avoid performing sequential finger movements during the time between the two experimental sessions.

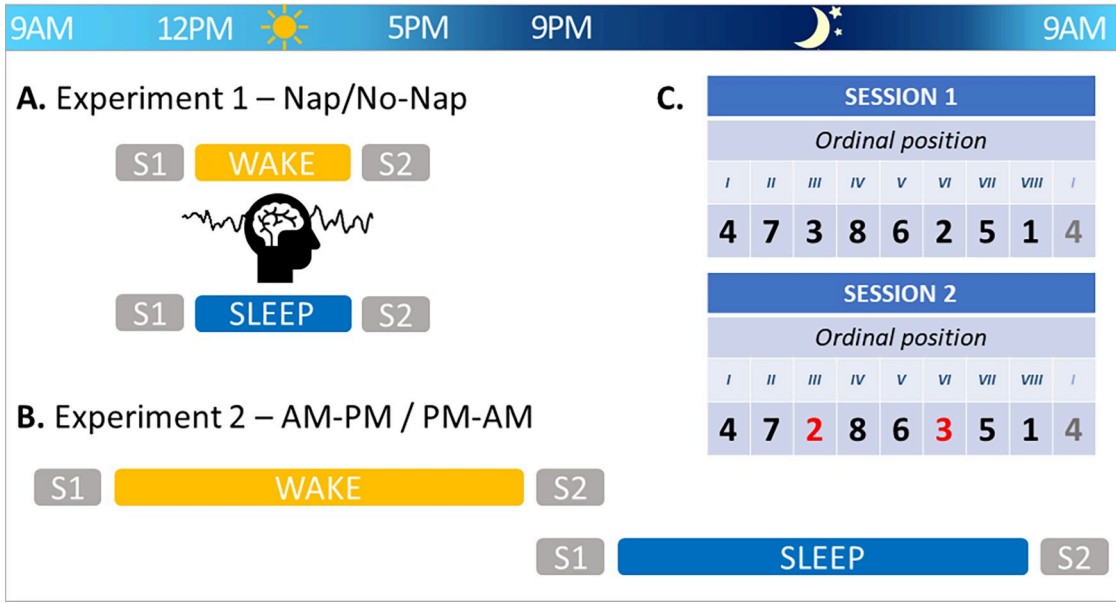

**Fig 1. Experimental design. A.** Experiment 1 consisted of a Nap/No-Nap design. Participants completed Session 1 (S1) in the morning, followed by a 90-minute episode of either sleep or quiet wakefulness monitored with standard polysomnography. They then completed Session 2 (S2) in the afternoon. **B.** Experiment 2 consisted of an AM-PM / PM-AM design. Participants completed S1 either in the morning or evening, and S2 on the same evening or on the following morning, respectively. **C.** For both experiments, during S1 and S2 participants performed a motor task (SRTT) during which they learned separate motor sequences (S1: Sequence 1, corresponding to keys 4-7-3-8-6-2-5-1, in which 1 through 8 are the left pinky to the right pinky fingers, excluding the thumbs; S2: Sequence 2, corresponding to keys 4-7-2-8-6-3-5-1). Sequence 2 was highly similar to Sequence 1, as 75% of the elements were in the same ordinal position. The positions of keys 2 and 3 (represented in red) were switched in Sequence 2, resulting in 4 novel key transitions to be learned relative to Sequence 1.

## Motor tasks

The motor task was identical to our earlier research [7]. Specifically, participants performed an explicit serial response time task (SRTT) coded and implemented in MATLAB ([28] MATLAB, 2020; Experiment 1) or Psytoolkit ([23, 24]; Experiment 2). The task consisted of an 8-choice response time task in which participants were instructed to react to cues shown on a screen. Eight squares, corresponding spatially to the 8 fingers used to perform the task (i.e., no thumbs) were presented on the screen. The color of the outline of the squares alternated between red and green, indicating periods of rest or practice, respectively. During practice, one of the squares was filled green, and participants were instructed to press the corresponding key with the corresponding finger as quickly and as accurately as possible. The cues either followed a pseudorandom sequence or a deterministic 8-element sequence. In the latter case, participants were explicitly informed that the order of key presses followed a sequential pattern, but were not given any additional information such as what the sequence was or how many keys it was composed of. For the pseudorandom condition, each of the eight keys/fingers appeared once in every run of 8 consecutive presses, ensuring that the distribution of keys/fingers was the same as the sequence condition. Each practice block contained 64 key presses (corresponding to eight repetitions of the eight-element sequence or eight consecutive random patterns of eight key presses) and was separated by 15-second rest periods.

During the first experimental session, participants completed 4 blocks of pseudorandom SRTT, designed to assess motor execution independent of a sequence to be learned. The pseudorandom SRTT was followed by the 24 blocks of sequential SRTT. The sequential runs for the first experimental session required participants to learn a specific key sequence (Sequence 1; 4-7-3-8-6-2-5-1, in which 1 through 8 are the left pinky to the right pinky fingers; see Fig 1C). During the second experimental session, participants performed a new motor sequence (Sequence 2; 24 blocks) in order to examine the integration of new movements into the preexisting motor sequence knowledge developed in Session 1. As the goal of the present study is to test whether post-learning sleep, as compared with wakefulness, is necessary for integration processes to take place, Sequence 2 was designed as per our previous research in order to optimize integration [7]. As in our previous research, the ordinal structure of Sequence 2 (4-7-2-8-6-3-5-1) was highly compatible with the previously learned sequence, as 75% of the elements were in the same ordinal positions as Sequence 1 (i.e., only elements 2 and 3 were in new ordinal positions; see Fig 1C). With this manipulation, 4 of the transitions in Sequence 2 were already learned in the previous session (i.e., 4–7, 8–6, 5–1, 1–4), and thus performance on these transitions reflects memory retention across the two experimental sessions. The other 4 transitions in Sequence 2 were not previously learned and thus were novel (i.e., 7–2, 2–8, 6–3, 3–5). Performance on these transitions then reflects integration of novel information into the cognitive-motor schema (corresponding to the ordinal structure of Sequence 1) acquired in the previous session [7]. The second experimental session ended with 4 blocks of the pseudorandom SRTT. Performance on these 4 blocks was compared to that on the 4 blocks of pseudo-random SRTT completed prior to sequence learning in Session 1 to assess across-sessions changes in task performance solely due to motoric improvement, i.e., unrelated to schema acquisition and integration. It should be noted that in both experimental sessions, the 24 blocks of the sequential SRTT were divided into two runs: a "training" run that included 20 practice blocks and a "test" run of 4 blocks. The test run was administered approximately 1 minute after the training run ended and was designed to assess end-of-training performance following the further dissipation of mental and physical fatigue [29].

For both pseudo-random and sequential variants of the SRTT, the primary outcome variables were: 1) response time (RT), i.e. time between cue presentation and subject key press,

and 2) accuracy. Individual trials were excluded from analyses when the measured response time was greater than 3 standard deviations above or below the participant's mean response time for that block. In Experiment 1 and 2, an average of 1.68% (Nap: 1.70%; No-Nap: 1.66%) and 1.71% (AM-PM: 1.72%; PM-AM: 1.70%) of trials were excluded from analyses, respectively. The averaged response times for correct movement transitions (i.e., both response n and n-1 were correct) and the percentage of correct transitions were computed for each block of practice on the task. For both measures and consistent with our previous work [7], averaging was done across all transitions (sessions 1 and 2) as well as on two subsets (Session 2 only): a) novel transitions (i.e., those transitions not performed in Session 1; 7–2, 2–8, 6–3 and 3–5); and b) learned transitions (i.e., those transitions that were also performed in Session 1; 4–7, 8–6, 5–1 and 1–4).

In each experimental session and following performance of the SRTT, participants performed a generation task in which they were asked to self-generate (i.e., without receiving visual cues) the sequence learned during that session. The purpose of the generation task was to assess sequence knowledge. The task consisted of 24 key presses, ideally corresponding to three 8-element sequences. Percentage of correct transitions and percentage of correct ordinal positions were extracted as the dependent measures of interest.

## Experiment 1

Approximately one week prior to the first SRTT session, subjects were invited for a habituation session during which they completed a nap monitored with polysomnography (PSG; acquisition details described below). This allowed participants to habituate to sleeping in our laboratory while wearing the PSG equipment.

Participants returned to the laboratory to complete two SRTT sessions in the same day, separated by a delay of approximately 4 hours (Fig 1A). During the delay between the two sessions, participants were afforded a 90-minute nap opportunity (Nap group) or watched a movie with minimal sound volume and no subtitles for the same duration (No-Nap group). These nap and quiet wakefulness episodes were both monitored with PSG. Thus, participants completed experimental Session 1 in the late morning (10am-1pm), the Nap or No-Nap conditions in the early afternoon (12–3:30pm) and Session 2 took place 30min after the Nap/No-Nap episode to ensure dissipation of sleep inertia (2pm-5:30pm).

Both habituation and experimental nap recordings were completed in a sound-dampening sleep laboratory, monitored with a digital sleep recorder (V-Amp, Brain Products, Gilching, Germany; bandwidth: DC to Nyquist frequency) and digitized at a sampling rate of 1000 Hz. Standard electroencephalographic (EEG) recordings were made from Fz, C3, Cz, C4, Pz, Oz, A1 and A2 according to the international 10–20 system. As the habituation session served to acclimate participants to sleep recordings and these data were not considered in formal analyses (other than sleep scoring), channels Fz, Pz and Oz were omitted to accelerate the procedure. A2 was used as the recording reference. A1 served as a supplemental individual EEG channel that could be used as a back-up reference if necessary. An electrode placed on the middle of the forehead served as the recording ground. Bipolar vertical and horizontal eye movements (electrooculogram: EOG) were recorded from electrodes placed above and below the right eye and on the outer canthus of both eyes, respectively. Bipolar submental electromyogram (EMG) recordings were made from the chin. Electrical noise was filtered using a 50 Hz notch.

Polysomnographic data from both the nap and no-nap intervals were scored using an automated algorithm [30]. The algorithm uses two EEG channels (C3 and C4, referenced to the right and left mastoids, respectively), which are then averaged, and two EOG channels (left and right EOG, referenced to the opposite side mastoids). The resulting three-channel PSG data

is then band-pass filtered between 0.3–45 Hz (EEG channel) or between 0.3–12 Hz (EOG channels), downsampled to 100Hz, and separated into 30s epochs. For each epoch, a fast Fourier transform is applied using a Hamming window of length 128, resulting in a spectrogram representing the time-frequency composition of the raw data. Each epoch is classified as belonging to a certain sleep stage according to a two-stage classification method: in the first stage, a deep convolutional neural network assigns sleep stage probabilities to the epoch based on its spectrogram; in the second stage, a multilayer perceptron generates revised sleep stage probabilities for the epoch by weighing the most probable sleep stage of the 5 preceding and succeeding epochs. The resulting most probable sleep stage is then selected as final score for that epoch. The automated sleep scores were then visually checked epoch by epoch by an experimenter. Where a mismatch arose between the experimenter's and the algorithm's score, the experimenter's score, determined according to standard criteria [31], was retained for the epoch.

For the nap sessions, in addition to sleep scoring, we extracted sleep features known to be involved in motor memory processes. Specifically, sleep spindles and slow waves were automatically detected in NREM stages 2 and 3 sleep epochs using the YASA python package [32]. The spindle detection algorithm, based on previous work by Lacourse and colleagues [33], uses three detection methods: first, a Short-Time Fourier Transform performed on consecutive 2 sec epochs, with 200 ms overlap, to estimate the relative power in the sigma frequency band over the broadband EEG signal. Second, a sliding window (300 ms, step size 100 ms) is used to compute the Pearson correlation coefficient between the broadband EEG data and the spindle frequency band (12–15 Hz) signal. Third, the same sliding window is used to compute the root mean squared (RMS) within the defined spindle frequency band. A spindle is detected when two of the three following conditions are met: the spindle band–broadband correlation exceeds 0.65; the relative spindle band power equals or exceeds 0.2; the moving RMS exceeds the mean RMS + 1.5 standard deviations. Events longer than 2 sec or shorter than 0.5 sec are discarded. Events closer than 500 ms are merged into single spindles. The slow wave detection algorithm, based on previous work by Massimini et al. [34] and Carrier et al. [35], uses a Finite Impulse Response filter with transition band of 0.2 Hz to apply a band-pass filter between 0.3 and 1.5 Hz to the raw EEG data. Events are defined as slow waves if they meet the following criteria: duration of the negative peak between 0.3 and 1.5 sec, duration of the positive peak between 0.1 and 1 sec, amplitude of the negative peak between 40 and 300 μV, amplitude of the positive peak between 10 and 200 μV. After detection of negative and positive peaks, the algorithm finds the closest positive peak per each negative peak and computes peak-to-peak amplitude (PTP). Only PTPs between 75 and 500 μV are retained for analysis. Density and amplitude of spindles and slow waves were computed separately per each midline channel Fz, Cz, and Pz and then averaged across channels (but see S5 Table for channel-level data). Spindle and slow wave density were computed as the total number of spindle or slow wave events detected divided by the total time spent in sleep stages N2 and N3, per participant.

## Experiment 2

The two experimental sessions were separated by a delay of approximately 12 hours (Fig 1B). Participants either performed the first session in the morning and the second in the evening of the same day (AM-PM group) or performed the first session in the evening and the second session in the morning of the following day (PM-AM group). All morning sessions were completed between 8.00 and 10.00 AM, and evening sessions between 8.00–10.00 PM.

During the delay between the two experimental sessions, participants followed their habitual daily schedule, which contained either a full night of sleep (PM-AM group) or 12h of continuous wake (AM-PM group). Participants of the PM-AM group were instructed to continue

to follow the regular sleep schedule (i.e., bed/wake time in a 2h window around their habitual schedule and minimum of 7h of sleep), while participants of the AM-PM group were instructed not to nap between the two sessions. Compliance to these instructions was assessed in the two groups with the St Mary Hospital questionnaire.

Experimental sessions were completed at the participants' own home, on their own computer, via a web-based platform [23, 24] that has been used in previous motor sequence learning research [36]. Prior to each experimental session, participants were instructed to remove all distractions from their environment such as computer or phone notifications, and to close the door to their room. To ensure participants' compliance to experimental instructions, subjects were visually monitored through a videocall service. Participants were required to keep both their sound and video active throughout the entire session.

## Statistical analyses

All analyses were performed using IBM SPSS Statistics for Windows, version 26 [37]. In case of violation of the sphericity assumption, we applied Greenhouse-Geisser corrections for $\varepsilon \leq 0.75$, Huynh-Feldt corrections for $\varepsilon > 0.75$ [38].

For each experiment, group differences in vigilance [25, 26] prior to the experimental sessions were assessed with separate *session* (2 levels) x *group* (2 levels) ANOVAs for each experiment. Sleep quantity and quality [27] for the night prior to the first motor task session were compared between groups with an independent-samples t-test for each experiment. Performance measures from the pseudo-random variant of the SRTT, reflecting general motor execution, were analyzed with separate *group* (2) x *block* (4) ANOVAS for each session. Performance on the sequential SRTT runs from the first session was assessed with *group* x *block* ANOVAs separately for each experiment, run (training (20 blocks) / test (4 blocks)) and dependent measure (RT / % correct). Last, performance on the generation task from Session 1 was analyzed with separate independent samples t-tests for the dependent measures of percentage of correct transitions generated and percentage of correct ordinal positions generated.

The primary, confirmatory data analyses aimed to test whether post-learning sleep resulted in enhanced memory integration compared with wakefulness (i.e., Nap vs. No Nap for Experiment 1 and PM-AM vs. AM-PM for Experiment 2). Separate two-way repeated-measures ANOVAs with the between-subject factor *group* and the within-subject factor *block* were conducted on the Session 2 sequential training and test runs to assess the effect of sleep on task performance. The analysis was performed on all movement transitions, as well as on two transitions subsets: a) novel transitions (investigating integration processes); and b) learned transitions (investigating memory retention processes). Similarly, two-way, repeated-measures ANOVAs with the between-subject factor *group* and the within-subject factor *transition type* (novel/learned) were conducted on response time and % correct transitions from the Session 2 post-training test run. This analysis aimed at comparing performance related to memory integration and retention between the two groups after extensive practice on the task (i.e., after the Session 2 training run; see [7]).

Additionally, for Experiment 1 only, we examined potential relationships between behavior and sleep features extracted from the nap period. Pearson correlations were performed between the extracted NREM sleep features (i.e., density and amplitude of spindles and slow waves) and a performance index (PI) on novel and learned transitions in Session 2. The PI is an aggregate speed-accuracy measure used extensively in previous work [39–41] and is computed with the following formula:

$$PI_x = exp^{-(SeqDur_x)} \times exp^{-(Errors_x)} \times 100 \tag{1}$$

Where $x$ = blocks of trials; SeqDur = sequence duration (i.e., reflecting movement speed); errors = proportion of incorrect transitions.

PI was averaged across the 20 training blocks of Session 2 and normalized by the individual's average PI on the 4 blocks of pseudorandom SRTT completed during Session 2 (i.e., reflecting motor execution on Session 2 independent of a sequence to be learned). As outlined in our pre-registration, this speed-accuracy measure was used solely for conducting correlations with relevant sleep features to reduce the number of correlations conducted (i.e., as opposed to correlating with speed and accuracy measures separately). For completeness, however, exploratory correlational analyses with separate speed and accuracy measures can be found in the supplement.

## Results

### Experiment 1: Effect of diurnal sleep

**Participant characteristics, vigilance and general motor performance.** Participant characteristics, sleep quality, quantity and duration for the night(s) preceding the experimental session, as well as subjective and objective measures of vigilance are provided in Table 1. Results from the corresponding statistical analyses can be found in the S1 Table. Performance on the pseudorandom SRTT, used to assess general motor execution prior to and following the sequential SRTT during sessions 1 and 2, respectively, is depicted in S1 Fig and results of the corresponding analysis are summarized S2 Table. In brief, and as expected, participants in the two experimental groups did not differ with regards to any of the reported measures (e.g., general characteristics, sleep prior to learning, vigilance and general motor execution).

**Sleep during Nap / wake interval.** A detailed characterization of sleep characteristics and features during the 90-minute nap opportunity is provided in Table 2. Briefly, participants in the nap group slept on average 58.0 ± 19.5 minutes and presented a sleep latency of 15.5 ± 14.0

**Table 1. Participant demographics and sleep and vigilance scores for each experimental group.**

| Variable | Experiment 1 | | Experiment 2 | |
|---|---|---|---|---|
| | Nap | No-Nap | AM-PM | PM-AM |
| $n$ | 25 | 25 | 29 | 28 |
| Female ($n$) | 17 | 17 | 21 | 19 |
| Age (years) | 22.8 (2.6) | 23.8 (2.9) | 22.9 (3.4) | 23.6 (3.0) |
| BAI score | 4.2 (4.4) | 3.2 (3.4) | 4.7 (3.5) | 3.8 (3.5) |
| BDI score | 4.7 (4.2) | 4.4 (3.8) | 5.2 (4.2) | 5.0 (4.6) |
| Handedness score | 87.0 (15.8) | 88.0 (11.4) | 84.0 (14.5) | 84.1 (13.7) |
| PSQI score | 3.3 (1.3) | 3.0 (1.6) | 2.9 (1.2) | 2.7 (1.4) |
| Daytime sleepiness score | 2.4 (2.1) | 3.7 (3.3) | 5.2 (3.1) | 5.1 (3.3) |
| Sleep duration, 3 nights prior to S2 (hours) | 8.8 (0.9) | 9.0 (0.8) | 8.4 (0.6) | 8.0 (0.7) |
| SMS duration (hours) | 8.0 (1.1) | 8.1 (0.9) | 7.8 (0.7) | 7.5 (0.7) |
| SMS quality | 4.0 (0.7) | 4.0 (0.7) | 4.1 (0.6) | 3.9 (0.7) |
| Session 1 SSS score | 2.5 (1.0) | 2.6 (1.1) | 2.3 (0.8) | 2.2 (0.9) |
| Session 2 SSS score | 2.5 (0.8) | 2.8 (1.4) | 2.4 (1.0) | 2.5 (0.7) |
| Session 1 PVT (ms) | 308 (32) | 303 (26) | 280 (29) | 291 (72) |
| Session 2 PVT (ms) | 310 (38) | 306 (28) | 276 (38) | 277 (50) |

Numbers represents the mean, with standard deviation in parentheses. BAI = Beck Anxiety Inventory [19]; BDI = Beck Depression Inventory [18]; PSQI = Pittsburgh Sleep Quality Index [20]; SMS = St. Mary's Hospital Sleep Questionnaire [27]; SSS = Stanford Sleepiness Scale [26]; PVT = psychomotor-vigilance task [25]. Nap, No-Nap: N = 25 in each group; AM-PM group: N = 29; PM-AM group: N = 27 for 'Sleep duration, 3 nights prior to S2', N = 28 for all else.

**Table 2. Average sleep characteristics for participants in the Nap group (Experiment 1).**

|  | Mean (Standard Deviation) |
| --- | --- |
| Total sleep duration (min) | 57.5 (19.9) |
| Non-REM duration (min) | 52.3 (16.3) |
| N1 duration (min) | 10.8 (8.8) |
| N2 duration (min) | 33.5 (14.6) |
| N3 duration (min) | 8.3 (10.2) |
| REM duration (min) | 5.1 (9.1) |
| Sleep latency (min) | 15.3 (12.7) |
| Efficiency (%) | 62.4 (21.8) |
| Wake after sleep onset (min) | 19.5 (18.0) |
| Spindle density (n/min) | 1.75 (1.22) |
| Spindle amplitude (μV) | 40.91 (10.03) |
| Slow wave density (n/min) | 3.68 (3.84) |
| Slow wave peak-to-peak amplitude (μV) | 121.64 (17.26) |

Sleep characteristics during the 90-minute nap opportunity for participants in the Nap group (Experiment 1). Spindle and slow wave characteristics were automatically detected in NREM sleep epochs using the YASA python package [32]. Further details are provided in the main text (Methods –Experiment 1). N = 25 for all measures except for duration of NREM3 and REM. Only 13 and 10 participants reached NREM3 and REM sleep, respectively.

minutes. All participants reached stage Non-REM 2 (N2) and maintained it for an average of 33.0 ± 14.5 minutes. Participants in the No-Nap group remained awake during the 90-minute interval, with an averaged sleep duration of 0.0 ± 0.5 minutes (maximum = 1.0 min).

**Motor task performance.** Performance on all transitions of the sequential SRTT is depicted in Fig 2 and the output of the corresponding statistical analyses is presented in Table 3.

In Session 1 of the sequential SRTT (i.e., prior to the experimental manipulation), participants successfully learned the motor sequence, as evidenced by the main effect of *block* for the dependent measure response time (Fig 2, Session 1; Table 3A). These improvements in speed did not come at the expense of decreased accuracy, as the percentage of correct transitions remained stable across practice blocks. As expected, there were no differences between the Nap and No-Nap groups during initial learning, as evidenced by the lack of significant group effects as well as *group x block* interactions for both speed and accuracy measures. A nearly identical pattern of results was observed in the short test administered immediately after the Session 1 training run. Namely, RT decreased across blocks whereas accuracy remained high and stable. Most importantly, there were no differences between groups for any of the performance measures. These results collectively demonstrate that motor learning did not statistically differ between groups prior to the experimental manipulation in Session 1.

To determine whether post-learning sleep facilitated performance on the new sequence in Session 2, we assessed group differences in behavioral performance on the post-sleep/wake SRTT sequence task (Fig 2, Session 2; Table 3A). Analogous to Session 1, a significant main effect of *block* on response time was observed, indicating successful learning of the second motor sequence, while accuracy remained high and stable. The two experimental groups did not differ on either RT or accuracy in the Session 2 training run, as evidenced by the lack of group and group by block effects. Similarly, there were no group differences in the short post-training test run for both dependent measures.

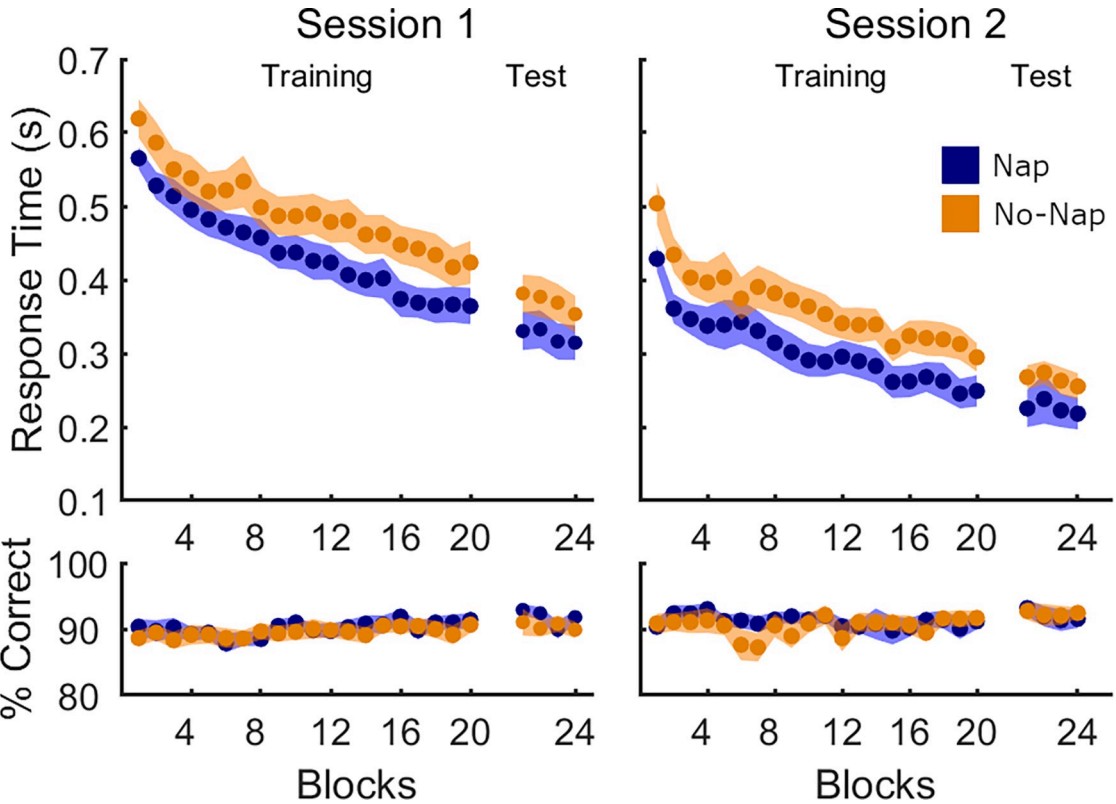

**Fig 2. Performance on all transitions in Experiment 1.** Mean response time (in seconds) and % correct transitions per block of sequence task (20 blocks of training, 4 blocks of test administered directly after training) are shown separately for the Nap group (blue, N = 25) and No-Nap group (orange, N = 25). The experimental manipulation (nap/no-nap) took place between Session 1 and Session 2. Shaded areas represent the standard error of the mean (SEM).

The sequence learned in Session 2 contained movement transitions that had been previously learned in Session 1 (learned transitions) as well as transitions that were new (novel transitions). We repeated our analyses on these two subsets of movement transitions separately to investigate memory retention and integration processes, respectively (Fig 3; Table 3B, 3C). Results were nearly identical to those obtained on the global analyses of all transitions detailed above. Significant block effects were observed for RT on both learned and novel transitions. Note there was a significant ($p$ = 0.048) effect of *block* on accuracy for novel transitions only, possibly indicating a decrease in performance difficulty with practice on the novel transitions during learning of Sequence 2. However, there were no significant differences between the two experimental groups in the training or test runs.

Last, repeated-measures ANOVAs with the between-subject factor *group* and the within-subject factor *transition type* (novel vs. learned) conducted on the Session 2 test revealed no group differences in performance on the two transition types after extensive task practice (*response time*–main effect *transition type*: $F(1,48) = 0.14$, $p = 0.71$, $\eta^2 = 0.003$; main effect *group*: $F(1,48) = 1.65$, $p = 0.21$, $\eta^2 = 0.033$; interaction: $F(1,48) = 0.13$, $p = 0.72$, $\eta^2 = 0.003$; *accuracy*–main effect *transition type*: $F(1,48) = 2.58$, $p = 0.12$, $\eta^2 = 0.051$; main effect *group*: $F(1,48) = 0.45$, $p = 0.51$, $\eta^2 = 0.009$; interaction: $F(1,48) = 0.04$, $p = 0.85$, $\eta^2 = 0.001$).

It is worth highlighting that the results presented above (see Table 3) did indicate trends for significant group differences in response time during the Session 2 training runs for all, learned and novel transitions (all $p = 0.07$). Specifically, and consistent with our hypotheses,

**Table 3. Sequential SRTT performance in Experiment 1.**

| Effect | RT | | | | Accuracy | | | |
|---|---|---|---|---|---|---|---|---|
| | df | F | p | Part η² | df | F | p | Part η² |
| **A. All Transitions** | | | | | | | | |
| *Session 1 Training* | | | | | | | | |
| Block | 5.79,277.7 | 78.44 | <0.001* | 0.620 | 12.2,583.8 | 1.19 | 0.29 | 0.024 |
| Group | 1,48 | 2.78 | 0.10 | 0.055 | 1,48 | 0.25 | 0.62 | 0.005 |
| B x G | 5.79,277.7 | 0.87 | 0.52 | 0.018 | 12.2,583.8 | 0.47 | 0.94 | 0.010 |
| *Session 1 Test* | | | | | | | | |
| Block | 2.19,105.0 | 6.10 | 0.002* | 0.113 | 2.65,127.3 | 0.94 | 0.41 | 0.019 |
| Group | 1,48 | 1.76 | 0.19 | 0.035 | 1,48 | 0.89 | 0.35 | 0.018 |
| B x G | 2.19,105.0 | 0.56 | 0.59 | 0.011 | 2.65,127.3 | 1.08 | 0.35 | 0.022 |
| *Session 2 Training* | | | | | | | | |
| Block | 5.46,262.1 | 42.1 | <0.001* | 0.467 | 10.4,500.6 | 1.4 | 0.17 | 0.028 |
| Group | 1,48 | 3.54 | 0.07 | 0.069 | 1,48 | 0.24 | 0.63 | 0.005 |
| B x G | 5.46,262.1 | 0.58 | 0.73 | 0.012 | 10.4,500.6 | 1.14 | 0.33 | 0.023 |
| *Session 2 Test* | | | | | | | | |
| Block | 1.75,80.4 | 2.56 | 0.09 | 0.053 | 3,138 | 0.77 | 0.51 | 0.016 |
| Group | 1,46 | 1.75 | 0.19 | 0.037 | 1,46 | 0.03 | 0.87 | 0.001 |
| B x G | 1.75,80.4 | 0.16 | 0.82 | 0.004 | 3,138 | 0.25 | 0.86 | 0.005 |
| **B. Learned Transitions** | | | | | | | | |
| *Session 2 Training* | | | | | | | | |
| Block | 5.94,284.9 | 27.04 | <0.001* | 0.360 | 9.78,469.2 | 1.75 | 0.07 | 0.035 |
| Group | 1,48 | 3.43 | 0.07 | 0.067 | 1,48 | 0.23 | 0.88 | <0.001 |
| B x G | 5.94,284.9 | 0.80 | 0.57 | 0.016 | 9.78,469.2 | 1.12 | 0.34 | 0.023 |
| *Session 2 Test* | | | | | | | | |
| Block | 1.98,91.1 | 1.16 | 0.32 | 0.025 | 3,138 | 0.71 | 0.55 | 0.015 |
| Group | 1,46 | 1.87 | 0.18 | 0.039 | 1,46 | 0.32 | 0.58 | 0.007 |
| B x G | 1.98,91.1 | 0.12 | 0.89 | 0.003 | 3,138 | 0.15 | 0.93 | 0.003 |
| **C. Novel Transitions** | | | | | | | | |
| *Session 2 Training* | | | | | | | | |
| Block | 6.01,288.5 | 44.70 | <0.001* | 0.482 | 10.3,494.3 | 1.85 | 0.048* | 0.037 |
| Group | 1,48 | 3.41 | 0.07 | 0.066 | 1,46 | 0.35 | 0.56 | 0.007 |
| B x G | 6.01,288.5 | 0.48 | 0.82 | 0.010 | 10.3,494.3 | 0.93 | 0.50 | 0.019 |
| *Session 2 Test* | | | | | | | | |
| Block | 1.91,88.0 | 3.42 | 0.04* | 0.069 | 3,138 | 0.57 | 0.64 | 0.012 |
| Group | 1,46 | 1.46 | 0.23 | 0.031 | 1,46 | 0.00 | 0.99 | <0.001 |
| B x G | 1.91,88.0 | 0.19 | 0.82 | 0.004 | 3,138 | 0.52 | 0.67 | 0.011 |

Response Time (RT) and Accuracy measures are provided for all transitions (**A**) during Sessions 1 and 2, and for learned transitions (**B**), and novel transitions (**C**) during Session 2 only. Significant values are marked with an asterisk. Df = degrees of freedom; Part η² = partial eta squared; B x G = block x group interaction. Sample size is of 24 participants per group for Session 2 test and 25 per group for all other analyses.

RT was faster for the Nap group as compared to No-Nap. A visual inspection of the data, however, suggests that at least a portion of this marginally significant difference can be explained by non-significant group differences during Session 1 (i.e., prior to the Nap / No-Nap experimental manipulation). To examine this possibility, we conducted exploratory, follow-up analyses on RT for the Session 2 training run but normalized to RT during the Session 1 test run. The normalization was computed for each individual by dividing the response time of each block of Session 2 by the response time averaged across the 4 blocks of Session 1 test. This

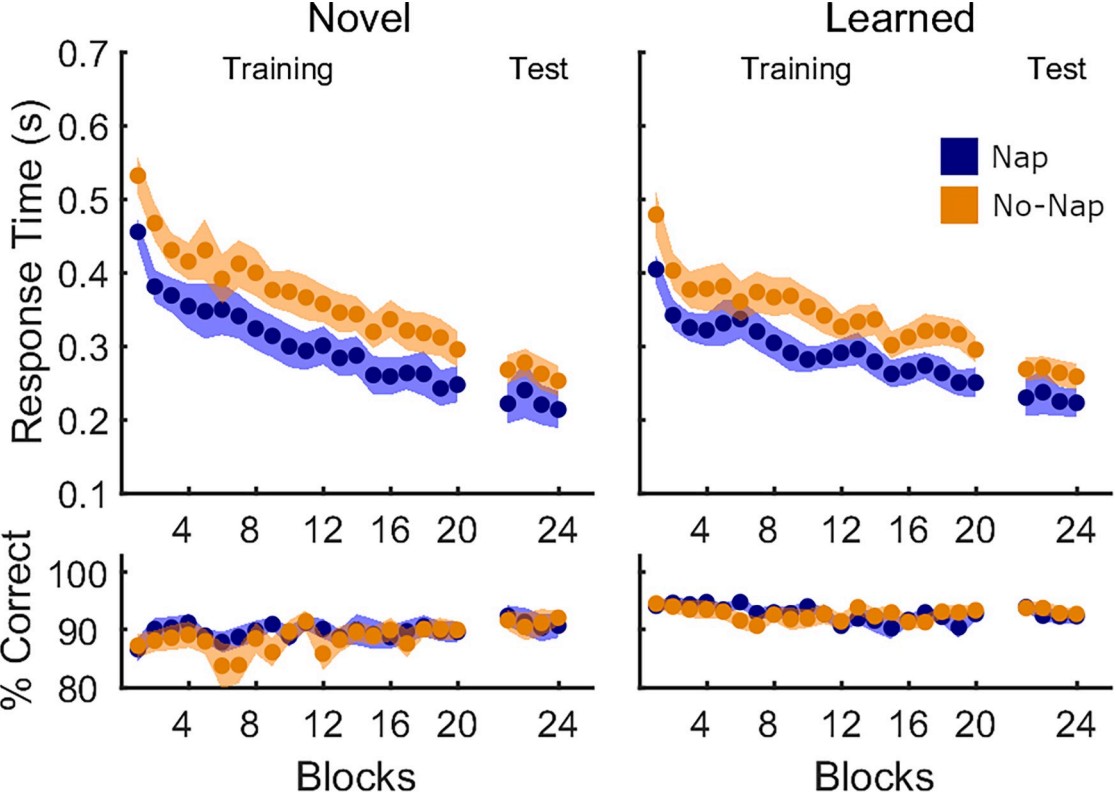

**Fig 3. Performance on novel and learned transitions in Session 2 of Experiment 1.** Mean response time (in seconds) and % correct transitions per block (20 blocks of training, 4 blocks of test administered directly after training) are shown separately for the Nap group (blue, N = 25) and No-Nap group (orange, N = 25). Shaded areas represent the SEM.

allowed to neutralize any effect on Session 2 data of spurious group differences in RT present prior to the experimental manipulation. These analyses revealed that after normalization, the trends for significant group differences in RT during Session 2 training were no longer present (see S2 Fig).

These results collectively demonstrate that a post-learning nap did not significantly influence schema-mediated motor memory consolidation processes.

**Sequence knowledge.** Sequence knowledge as measured by the generation task (percentage correct transitions, percentage correct ordinal positions) did not differ between groups in either session (see S3 Table).

*Correlations between sleep and behavior.* Per our pre-registration, we examined the relationship between performance in the post-nap sequential SRTT session and sleep features during the nap period. Pearson correlations were performed between extracted NREM sleep features of interest (i.e., duration of NREM sleep, spindle density, spindle amplitude, slow wave density, slow wave amplitude) and the speed-accuracy aggregate measure performance index (PI). [For completeness, PI for all, learned and new transitions is depicted in S3 Fig and corresponding statistical analyses are in S4 Table.] We found no significant correlations between sleep characteristics and PI during Session 2 (S6A Table). As exploratory analyses, we also conducted correlations between sleep characteristics and the behavioral measures of response time and accuracy. These analyses also revealed no significant correlations (S6B and S6C Table). We additionally present analogous correlation analyses performed on single-channel data in S7–S9 Tables, with no significant results surviving comparison for multiple corrections.

These results indicate that there were no significant relationships between sleep characteristics in the 90-minute nap opportunity and behavioral measures reflecting memory integration or retention processes.

### Experiment 2: Effect of nocturnal sleep

**Participant characteristics, vigilance and general motor performance.** Participant characteristics, sleep quality and quantity prior to the experimental session, and vigilance scores did not differ between groups (see Table 1 for group means and S10 Table for corresponding statistical output). Furthermore, we observed no group differences in performance in the pseudo-random SRTT administered prior to and following the sequential SRTT (see S4 Fig and S11 Table). These results indicate that the two groups did not statistically differ in any of the confounding factors assessed.

**Motor task performance.** Performance on all transitions of the sequential SRTT is depicted in Fig 4 and the output of the corresponding statistical analyses is presented in Table 4. Response time during performance of the Session 1 sequential SRTT (Fig 4, Session 1; Table 4A) significantly improved across blocks, demonstrating successful learning of the motor sequence, but did not differ between the two experimental groups. An analogous ANOVA conducted on the Session 1 post-training test detected no significant effect of *block* nor group differences in response time, indicating that both groups reached a stable and comparable performance post-training. Accuracy remained high and stable during the training run, but there was a significant effect of block on accuracy in Session 1 test (all transitions),

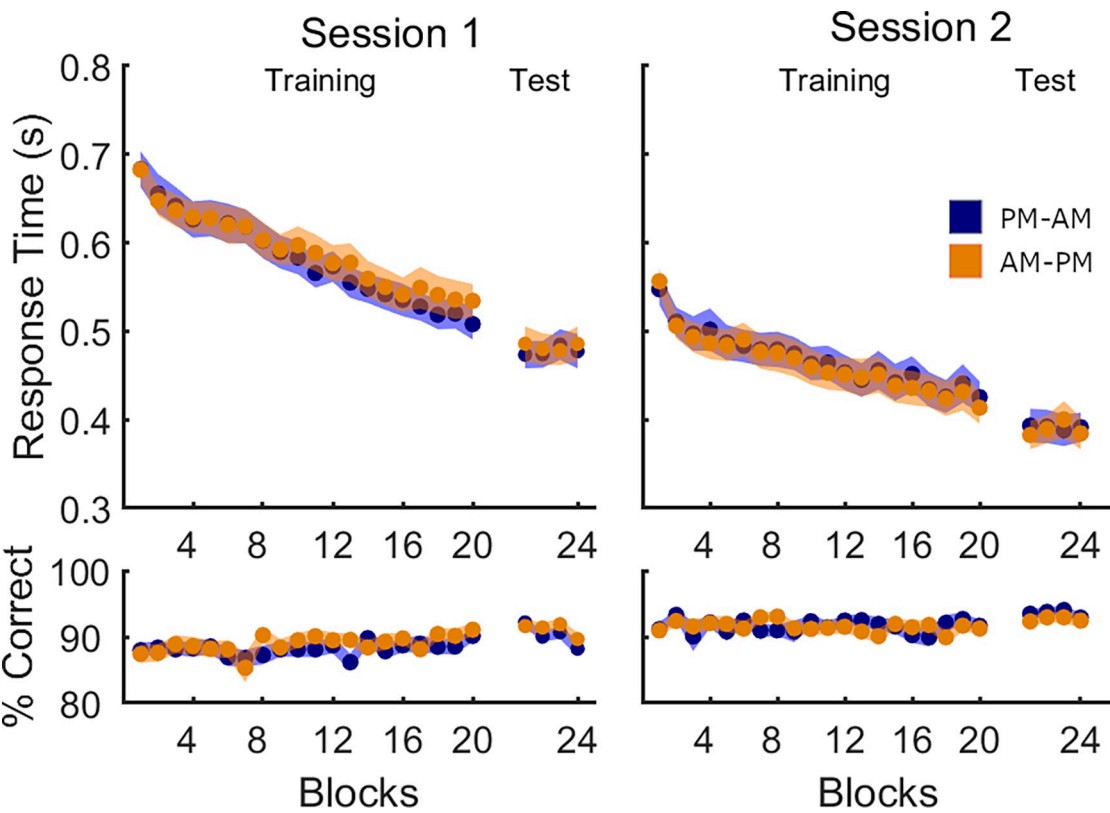

**Fig 4. Performance on all transitions in Experiment 2.** Mean response time (in seconds) and % correct transitions per block of sequence task are shown separately for the PM-AM (blue, N = 28) and AM-PM (orange, N = 29) groups. Shaded areas represent the SEM.

**Table 4. Sequential SRTT performance in Experiment 2.**

| Effect | RT | | | | Accuracy | | | |
|---|---|---|---|---|---|---|---|---|
| | df | F | p | Part $\eta^2$ | df | F | p | Part $\eta^2$ |
| **A. All Transitions** | | | | | | | | |
| *Session 1 Training* | | | | | | | | |
| Block | 6.3,347.4 | 77.67 | <0.001* | 0.585 | 11.9,655.3 | 1.61 | 0.09 | 0.028 |
| Group | 1,55 | 0.11 | 0.75 | 0.002 | 1,55 | 0.56 | 0.46 | 0.010 |
| B x G | 6.3,347.4 | 1.00 | 0.43 | 0.018 | 11.9,655.3 | 0.87 | 0.58 | 0.016 |
| *Session 1 Test* | | | | | | | | |
| Block | 3,165 | 0.37 | 0.78 | 0.007 | 2.8,154.4 | 4.70 | <0.01* | 0.079 |
| Group | 1,55 | 0.04 | 0.84 | 0.001 | 1,55 | 1.11 | 0.30 | 0.020 |
| B x G | 3,165 | 1.18 | 0.32 | 0.021 | 2.8,154.4 | 0.65 | 0.58 | 0.012 |
| *Session 2 Training* | | | | | | | | |
| Block | 8.9,489.3 | 40.46 | <0.001* | 0.424 | 11.4,626.9 | 0.69 | 0.76 | 0.012 |
| Group | 1,55 | 0.03 | 0.86 | 0.001 | 1,55 | 0.05 | 0.94 | <0.001 |
| B x G | 8.9,489.3 | 0.40 | 0.94 | 0.007 | 11.4,626.9 | 1.02 | 0.43 | 0.018 |
| *Session 2 Test* | | | | | | | | |
| Block | 3,165 | 0.55 | 0.65 | 0.010 | 3,165 | 0.49 | 0.69 | 0.009 |
| Group | 1,55 | 0.01 | 0.93 | <0.001 | 1,55 | 1.63 | 0.21 | 0.029 |
| B x G | 3,165 | 1.59 | 0.19 | 0.028 | 3,165 | 0.08 | 0.97 | 0.002 |
| **B. Learned Transitions** | | | | | | | | |
| *Session 2 Training* | | | | | | | | |
| Block | 9.1,501.1 | 22.48 | <0.001* | 0.290 | 10.5,578.7 | 2.04 | 0.03* | 0.036 |
| Group | 1,55 | 0.10 | 0.75 | 0.002 | 1,55 | 0.29 | 0.59 | 0.005 |
| B x G | 9.1,501.1 | 0.64 | 0.77 | 0.011 | 10.5,578.7 | 0.87 | 0.56 | 0.016 |
| *Session 2 Test* | | | | | | | | |
| Block | 3,165 | 0.41 | 0.75 | 0.007 | 3,165 | 0.88 | 0.45 | 0.016 |
| Group | 1,55 | 0.09 | 0.77 | 0.002 | 1,55 | 1.13 | 0.29 | 0.020 |
| B x G | 3,165 | 1.59 | 0.19 | 0.028 | 3,165 | 0.17 | 0.91 | 0.003 |
| **C. Novel Transitions** | | | | | | | | |
| *Session 2 Training* | | | | | | | | |
| Block | 9.2,503.1 | 46.49 | <0.001* | 0.458 | 12.7,695.7 | 0.72 | 0.74 | 0.013 |
| Group | 1,55 | 0.001 | 0.97 | <0.001 | 1,55 | 0.28 | 0.60 | 0.005 |
| B x G | 9.2,503.1 | 0.40 | 0.94 | 0.007 | 12.7,695.7 | 1.24 | 0.25 | 0.022 |
| *Session 2 Test* | | | | | | | | |
| Block | 3,165 | 0.51 | 0.68 | 0.009 | 3,165 | 0.40 | 0.75 | 0.007 |
| Group | 1,55 | 0.01 | 0.91 | <0.001 | 1,55 | 1.33 | 0.25 | 0.024 |
| B x G | 3,165 | 1.29 | 0.18 | 0.023 | 3,165 | 0.32 | 0.81 | 0.006 |

Response Time (RT) and Accuracy measures are provided for all transitions (**A**) during Sessions 1 and 2, and for learned transitions (**B**), and novel transitions (**C**) during Session 2 only. Significant values are marked with an asterisk. Df = degrees of freedom; Part $\eta^2$ = partial eta squared; B x G = block x group interaction. Sample size for all analyses is of 29 participants for the AM-PM group and 28 for the PM-AM group.

caused by decreased accuracy on the last test block (average 89%, compared with 91% in the first three blocks). However, there were no differences in accuracy across groups. These results also collectively indicate that there were no time-of-day effects on performance in Session 1.

Using analogous analyses to those performed on Experiment 1, we compared post-sleep/ wake performance to determine whether nocturnal sleep facilitated learning of the new motor sequence. During the Session 2 training run, we observed no significant group differences in either response time or accuracy, nor were there significant *block × group* interactions (see

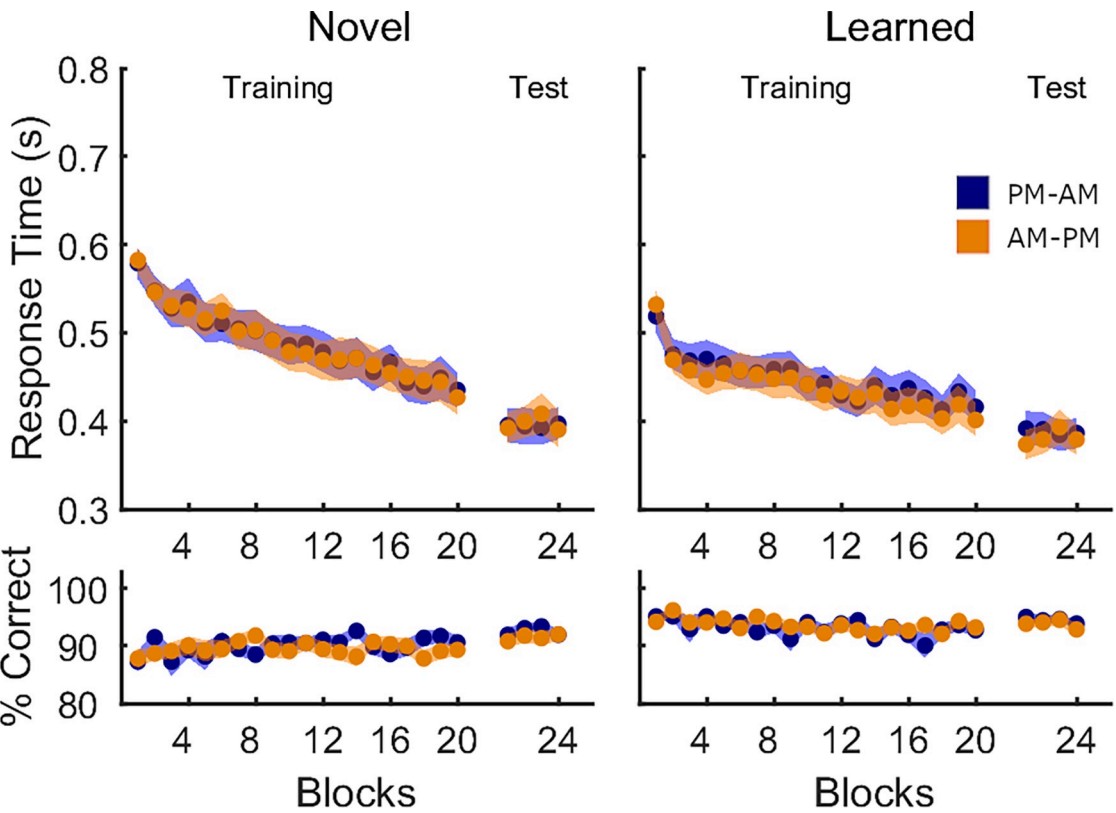

**Fig 5. Performance on novel and learned transitions in Experiment 2, Session 2.** Mean response time (in seconds) and % correct transitions per block of sequence task are shown separately for the PM-AM (blue, N = 28) and AM-PM (orange, N = 29) groups. Shaded areas represent the SEM.

Fig 4, Session 2). Response time significantly improved across blocks, an effect that was not observed for accuracy. Similar analyses repeated on the test run revealed no effect of *block*, *group*, or their interaction for both response time and accuracy. We repeated analyses on the two subsets of learned and novel movement transitions and again observed no group differences (Fig 5, see Table 4B, 4C for detailed statistics). Collectively, these results suggest that the two experimental groups successfully learned the second motor sequence, with no differences in performance between groups.

The *group* x *transition type* (novel vs. learned) ANOVA conducted on the Session 2 test revealed no group differences, but a trend for a significant effect of *transition type* on response time (main effect *transition type*: $F(1,55) = 3.19$, $p = 0.08$, $\eta^2 = 0.055$; main effect *group*: $F(1,55) = 0.01$, $p = 0.931$, $\eta^2 < 0.001$; interaction: $F(1,55) = 0.61$, $p = 0.44$, $\eta^2 = 0.011$) and a highly significant effect of *transition type* on accuracy (main effect *transition type*: $F(1,55) = 26.24$, $p < 0.001$, $\eta^2 = 0.323$; main effect *group*: $F(1,55) = 1.64$, $p = 0.21$, $\eta^2 = 0.029$; interaction: $F(1,55) = 0.25$, $p = 0.62$, $\eta^2 = 0.005$). These effects were driven by reduced response time and increased accuracy for learned transitions (RT–*learned*: 385 ± 90ms, *novel*: 396 ± 99ms; accuracy–*learned*: 94 ± 2%, *novel*: 92 ± 4%).

Consistent with Experiment 1, the results of the second experiment did not reveal any beneficial effect of sleep on the integration of novel motor information into a pre-existing schema.

**Sequence knowledge.** Results of the generation task indicated that sequence knowledge (percentage correct transitions; percentage correct ordinal positions) was comparable between groups during both sessions (see S12 Table for details).

## Discussion

The current study investigated whether diurnal (Experiment 1) or nocturnal (Experiment 2) sleep following the acquisition of a motor schema enhances the integration of novel, schema-compatible motor information. To our knowledge, this is the first study to examine the role of sleep in schema-mediated integration in the motor memory domain in humans (but see [42] for an investigation in mice). Our results failed to reject the null hypothesis that sleep provides no beneficial effect in schema-mediated integration, as demonstrated by the lack of group differences in post-sleep motor performance. Furthermore, we detected no significant correlations between NREM sleep features (i.e., density and amplitude of spindles and slow waves) and post-sleep behavioral performance. Collectively, our results suggest that integration of novel motor information into a cognitive-motor schema may not specifically benefit from post-learning sleep.

Previous literature in the declarative memory domain suggested a beneficial role of sleep in both declarative schema formation and schema-mediated integration [1, 11–14], although a recent investigation reported no such beneficial effect of sleep for schema-congruent memories [16]. Current models of the beneficial effect of sleep on schema formation and integration posit that overlapping memory replay during sleep would lead to the formation of cognitive schemas stored in the neocortex. Specifically, offline replay of memory engrams containing partially overlapping features could lead to a strengthening of these shared features across memories and to a simultaneous weakening of the non-shared features during global synaptic downscaling [1]. While this memory replay is initiated by the hippocampus, only the synaptic connections between neocortical neurons which are shared between memories would be highly potentiated and able to survive this downscaling. Thus, when new information that shares features with an established cognitive schema is presented, simultaneous activation of schema-related neocortical neurons and of the hippocampal neurons representing the new memory may then lead to the fast assimilation of the new memory into the schema and its fast transfer to neocortical storage sites [1]. In summary, sleep is suggested to play a crucial role in schema formation and integration by providing an optimal environment for replay and strengthening of relevant memory features, and forgetting of irrelevant ones.

In contrast to earlier and aforementioned work in the declarative domain, the results of the current study do not afford the conclusion that that sleep facilitates the integration of new *motor* information into a compatible schema. It is thus tempting to speculate that sleep benefits schema-facilitated integration in the declarative but not in the motor memory domain. However, such an explanation would not be consistent with research accumulated over the last 15 years which has demonstrated that the consolidation of memories formed during motor tasks share common features and rely on similar neural substrates and networks as the consolidation of memories formed during declarative tasks. Specifically, neuroimaging studies have illustrated the involvement of the hippocampal formation in motor sequence learning [43–48] and the involvement of the striatum in certain hippocampal-dependent tasks [49–51]. The declarative and procedural systems were additionally demonstrated to interact directly, with learning of declarative task interfering with learning of procedural tasks and vice versa [52–54]. Furthermore, the consolidation of both declarative and motor memories has been shown to be enhanced by sleep [1, 55–57], and by specific sleep features in particular, including sleep spindles [12, 58, 59] and slow oscillations [11, 60, 61]. Consequently, while the precise mechanisms underlying schema formation and integration in motor memory are currently unknown, it is reasonable to assume that this process will exhibit commonalities to that proposed in the declarative memory. In this light, the results of our current research were unexpected.

It could be speculated that the null effect on motor memory integration in the current study is due to the duration between experimental sessions (i.e., 4 and 12 hours in Experiments 1 and 2, respectively). Specifically, it is possible that longer time intervals between acquisition and integration sessions are necessary for the beneficial effect of sleep to develop. Our previous behavioral experiment [7] employed a delay of 24 hours between schema acquisition and integration. In the current study, we decided to limit the time delay to 4 hours for the diurnal sleep study and 12 hours for the nocturnal sleep study. This choice was motivated by the inherent difficulties associated with administering a sleep/wake intervention over a 24-hour period without causing sleep deprivation, which would introduce additional potential confounding factors. One previous study in the declarative memory domain which employed an AM-PM/PM-AM design comparable to our second experiment similarly observed no influence of sleep on schema integration at short delays (up to 10 hours post-learning); however, post-learning sleep significantly enhanced schema integration at a one-year retest [62]. Thus, it is possible that post-encoding sleep initiates a process of long-term motor schema retention which is only appreciable after longer intervals of time. This is a possibility that we could not test with our current design, but may be explored in future research.

In addition to finding no group differences in the integration of new motor information into a compatible schema, we similarly found no effect of sleep on retention of previously learned information (i.e., performance on the learned transitions). This result is not consistent with extensive literature detailing a beneficial effect of sleep on memory consolidation and retention processes [10]. However, it is worth noting that our experimental design is distinct from that generally employed in memory retention studies, as previously learned information is here embedded in a novel context. Accordingly, our design is optimized to test integration of novel information into an existing schema and the impact on the assessment of previously learned motor information is unclear.

The current research does suffer from a few limitations. First, the lack of additional experimental groups tested with a schema-*incompatible* framework makes it difficult to conclusively determine whether our experimental groups display a schema effect. We did not include schema-incompatible groups in our design, as our research question aimed to investigate the influence of sleep on schema-facilitated integration. It is important to note, however, that our original publication demonstrated a schema-compatibility effect across two independent samples. Second, and specific to Experiment 2, our assessment of overnight sleep in the PM-AM group and of daytime (lack of) sleep in the AM-PM group depended solely on self-report questionnaires. Contact with participants was limited as much as possible due to health and safety regulations at the time of data collection. Thus, it was not possible to conduct the experiment entirely in the laboratory or to obtain objective measures of activity through worn devices such as the actigraphy monitor employed in Experiment 1. However, it should be noted that the self-report questionnaire we used (St. Mary's sleep questionnaire) has been amply utilized in previous research as an effective subjective measure of sleep quantity and quality [63–68]. Finally, while our sample size for the current experiments was computed based on previous work employing the same motor task [7], it is worth explicitly stating that we were powered to detect a relatively large effect (i.e., Cohen's f = 0.383). Accordingly, it is feasible that statistical power was not sufficient to detect smaller effects. Nonetheless, it should be emphasized that the sizes of our experimental groups are larger than the vast majority of sleep and motor memory research [69] and that our results were consistent across two independent experiments.

In conclusion, we tested whether sleep influences the integration of novel motor information into a previously acquired cognitive-motor schema. Results of behavioral analyses failed to provide evidence for a performance benefit for participants who were allowed to sleep post-learning compared with those who remained awake. Furthermore, we found no significant

relationship between sleep characteristics and post-sleep performance. These results suggest that sleep does not enhance schema integration in the motor memory domain. Future research may explore whether post-learning sleep enhances motor memory integration after longer (i.e., > 24 hours) retention delays.

## Supporting information

**S1 Fig. Performance on the pseudo-random SRTT in Experiment 1.**
(PDF)

**S2 Fig. Normalized response time during Session 2 training and test runs.**
(PDF)

**S3 Fig. Aggregate speed-accuracy measure performance index (PI) during session 2 of Experiment 1.**
(PDF)

**S4 Fig. Performance on the pseudo-random SRTT in Experiment 2.**
(PDF)

**S1 Table. Group differences in participant characteristics and assessments of sleep and vigilance for Experiment 1.**
(PDF)

**S2 Table. Performance on the pseudo-random SRTT in Experiment 1.**
(PDF)

**S3 Table. Performance on the generation task in Experiment 1.**
(PDF)

**S4 Table. Output of statistical analyses performed on PI in Session 2 of Experiment 1.**
(PDF)

**S5 Table. Average sleep characteristics for participants in the Nap group (Experiment 1), per each midline channel.**
(PDF)

**S6 Table. Correlations between sleep features (Nap group only) and performance in the sequential SRT task during Session 2 for Experiment 1.**
(PDF)

**S7 Table. Correlations between sleep features (Nap group only) and performance in the sequential SRT task during Session 2 for Experiment 1, channel Fz.**
(PDF)

**S8 Table. Correlations between sleep features (Nap group only) and performance in the sequential SRT task during Session 2 for Experiment 1, channel Cz.**
(PDF)

**S9 Table. Correlations between sleep features (Nap group only) and performance in the sequential SRT task during Session 2 for Experiment 1, channel Pz.**
(PDF)

**S10 Table. Participant characteristics and assessments of sleep and vigilance for Experiment 2.**
(PDF)

**S11 Table. Performance on pseudo-random SRTT in Experiment 2.**
(PDF)

**S12 Table. Performance on the generation task in Experiment 2.**
(PDF)

## Acknowledgments

We wish to thank the participants for their time and effort devoted to this study.

## Author Contributions

**Conceptualization:** Genevieve Albouy, Bradley R. King.

**Formal analysis:** Serena Reverberi.

**Funding acquisition:** Genevieve Albouy, Bradley R. King.

**Investigation:** Serena Reverberi, Nina Dolfen.

**Methodology:** Bradley R. King.

**Software:** Anke Van Roy.

**Supervision:** Nina Dolfen, Genevieve Albouy, Bradley R. King.

**Visualization:** Serena Reverberi.

**Writing – original draft:** Serena Reverberi.

**Writing – review & editing:** Serena Reverberi, Nina Dolfen, Anke Van Roy, Genevieve Albouy, Bradley R. King.

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
