## [Decision Letter · Decision Letter 0]

13 Oct 2022

PONE-D-22-16102Sleep does not influence schema-facilitated motor memory consolidation

PLOS ONE

Dear Dr. ALBOUY,

Thank you for submitting your manuscript to PLOS ONE. After careful consideration, we feel that it has merit but does not fully meet PLOS ONE’s publication criteria as it currently stands. Therefore, we invite you to submit a revised version of the manuscript that addresses the points raised during the review process.

Three reviews have been provided, with Reviewer 1 providing the most detailed comments. These all appear sensible to me with one suggestion meriting an additional comment. Reviewer 1 mentions carrying out correlations with REM sleep. I agree that these would be well motivated, although as with the main analysis these would have limited statistical power to detect effects unless they were very large. Obviously these were not pre-registered and if carried out then they should be flagged as such, and I don't feel you should change your introduction to accommodate them, and instead could be added as exploratory analyses in your results.

In my view, the submission does not meet the journal's requirements for the "minimal data set" availability. Although data was provided to recreate the graphs, data is not currently available to replicate the study findings. While providing data to produce graphs is described as an example on the data availability policy description (https://journals.plos.org/plosone/s/data-availability), this should be interpreted in the context of a multi-discipline journal with discipline specific practices in regard to analysis and sharing. For psychology, the standard practice would be to make subject level available that provided the inputs to the analysis, though ideal best practice may be to include trial level data. Therefore I would ask for subject level data to be made available to allow these analyses to be replicated or alternative analyses performed.

I make some specific points about power below but I would describe a major limitation of this work that was not mentioned is that it was powered to detect large effects. Reviewer 1 suggests it had "adequate statistical power" yet this would only apply to large effects. This means it had low power to detect smaller effects, which might be reflective of typical effects sizes in psychological research, and increase the chances of results reflecting Type 2 error in this case. And it would seem smaller effects (e.g. d = .4) would be still taken as theoretically informative as to the effects of sleep had they been found and sufficient power was available to make them statistically significant. This is an issue for both the performance data and the correlation data. This is reflects a general problem of too optimistic effect size estimates as the basis of power analyses, given we know that published effect sizes will likely be over estimates. While the sample size is typical or even higher than many sleep and memory studies in the past, a key issue for the replication crisis has been the use of small samples (e.g. n < 30 per group) in between-subjects designs. In the case of sleep research, while there are difficulties in obtaining large sample sizes continued use of sample small sample sizes will have negative consequences for the credibility of the field (see Cordi & Rasch, 2021) and increasingly replication studies in the field sleep research are showing challenges here. In this particular case, n = 25 per group would lead to any effect size of Cohen's d < .57 as being non-significant (in a between-subjects t-test). For instance, I calculated the key effect in Table C as a Cohen's d = .51, which was not significant (p = .07). Given estimates that the average effect size in psychology might be around d = .4, the design had poor sensitivity to detect effects around this magnitude. However, the language of the discussion was very categorical in wording such as "no evidence for an effect of sleep," and "no performance benefit", which is too strong, particularly given Null Hypothesis Significance Testing framework employed. More formally, it appears the paper accepts the null hypothesis rather than failing to reject it. 

I did find the post-hoc analyses to suggest the relatively large (but not significant) effect in the first experiment seemed due to baseline differences reasonably persuasive, although I believe this was not pre-registered. And the consistency of the pattern of results across both studies is suggestive of a null effect. But if true power was low (i.e. there was a real but moderate effect), then two null effects in a row being a result of Type 2 error should be taken seriously as a candidate explanation for these pattern of results. The general point remains that with small sample sizes it is harder to make strong conclusions about the absence of moderate but realistic effects, particularly given the confidence intervals of effect sizes estimate with small sample studies will be include a large range of plausible effect sizes. 

PLOS One is a journal that does welcome null results, and I think these results despite the small sample sizes are worth publishing, but in-line with our editorial policy conclusions should be calibrated to what the data can support. Therefore I would like to see a more measured discussion of these findings in a revision which gives greater weight to the uncertainty present, and gives some credence to alternative explanations for these results in the light of effect sizes and statistical power. 

We look forward to receiving your revised manuscript.

Kind regards,

Shane Lindsay

Academic Editor

PLOS ONE

Journal Requirements:

**Additional Editor Comments:**

1. The power analysis states the "primary effect of interest in this earlier work was the group difference in performance on the novel transitions learned in the Session 2 training phase, reflecting integration processes."

It would be helpful to elaborate in his section more on the exact statistical method and contrast that represents the primary effect of interest (i.e. an ANOVA, a t-test etc...), including the dependent measure(s). One issue is "performance" is underspecified, given the results appear to use both RT and errors as performance measures. I believe it may mean RT in this context but it should be stated explicitly if RT was the primary test of your hypothesis. This could also be addressed in the introduction in reference to the previous work - did it find significant effects with RT and with accuracy or just RT? If the latter, then I presume the hypotheses here were focused on the RT effect, and this should be clarified. 

2. "This group difference presented a Cohen's d effect size of 0.86."

On my examination of King et al. (2019) it was not straightforward to identify where this Cohen's d came from partly because the exact test is underspecified (see above point). But it appeared that there was no direct comparison between a 24 period consolidation group in Experiment 1a, and an intermediate group in Experiment

Addressing the above point may help here but it should be made more clear how this value was determined, which made some clarification on how the design of the studies differed. In fact, from my understanding it seems misleading to state the "primary effect of interest in this earlier work was the group difference" if there no was test of the group difference, thought this could result from ambiguity in the wording of this statement. It seems instead that the study involved a significant effect in Experiment 1a, and a non-significant effect in Experiment 2, but without direct comparisons one cannot conclude they are different (e.g. see Gelman & Stern, 2012). This should be highlighted as otherwise the impression could be misleading. 

3. Trying to make sense of this in the present work, it seems like it could refer to the analysis on p. 22 "Last, repeated-measures ANOVAs with the between-subject factor group and the within-subject factor transition type (novel vs. learned) conducted on the Session 2 test revealed no group differences in performance on the two transition types after extensive task practice". Yet this would not be a test of performance just on novel transitions, as it includes novel vs. learned as a factor in an ANOVA, rather than just a between-subjects comparison on RT for novel transitions in Session 2 training. But Table 3 section C does appear to show this contrast referred to in the participants section as the key effect, with the group difference of F(1, 48) = 3.31, p = .07. However, this does not seem to match what is described in the text. Could this be clarified in the manuscript. 

4. The design of the study sets up a difference in memory integration for the novel sequences, compared with memory retention for the learned sequences. However, the discussion seems exclusively focused on integration (e.g. wording of first paragraph of the discussion). While I understand it might be the primary focus, it may be useful to at least mention for retention as well in the discussion section as it seemed to be important to the design and a worthwhile question on its own, especially if you might expect retention to be a prerequisite for integration. 

5. "In conclusion, we tested whether sleep is necessary for fast integration of novel motor information into a previously acquired cognitive-motor schema. " I struggled with this sentence as it doesn't appear this is what was tested, and what such a test would entail. What was actually tested was whether sleep lead to better performance compared with a wake group on a range of measures. Instead, a test for if sleep is "necessary" might take the form of some kind of test to show that a group without sleep were not able to show "fast integration". But the results all appeared to show group contrasts. This seems like the framing could be more relevant to the design of the King et al. (i.e. the compatible vs non-compatible contrast) study rather than this study. This should be addressed. 

**Reviewers' comments:**

**Reviewer's Responses to Questions**

**Comments to the Author**

1. Is the manuscript technically sound, and do the data support the conclusions?

Reviewer #1: Yes

Reviewer #2: Yes

Reviewer #3: Yes

2. Has the statistical analysis been performed appropriately and rigorously? 

Reviewer #1: Yes

Reviewer #2: Yes

Reviewer #3: Yes

3. Have the authors made all data underlying the findings in their manuscript fully available?

Reviewer #1: Yes

Reviewer #2: No

Reviewer #3: Yes

4. Is the manuscript presented in an intelligible fashion and written in standard English?

Reviewer #1: Yes

Reviewer #2: Yes

Reviewer #3: Yes

5. Review Comments to the Author

Reviewer #1: Albouy and colleagues investigate the role of diurnal (Experiment 1) and nocturnal (Experiment 2) sleep in schema-mediated motor sequence memory consolidation. The participants learned a motor sequence task which was followed by either sleep or an equivalent period of wake (90-min or 12 h, depending on the experimental paradigm). Next, learning of a second, highly compatible, sequence took place. The authors report no difference in behavioural performance on the new sequence between the sleep and wake group in either experiment, suggesting that the integration of novel motor information into a pre-existing schema does not benefit from post-learning sleep.

The data collection and analyses plan for Experiment 1 was pre-registered, giving more confidence in the tests conducted. Experiment 2 was exploratory but the data analysis followed that of the pre-registered Experiment 1. Furthermore, the sample sizes for both experiments were determined on the basis of the effect size measured in a similar study, ensuring adequate statistical power for each experiment. The experiments are also well controlled, with a habituation night for Experiment 1 and wake conditions for each experiment.

Overall, this is an interesting and well-controlled piece of work that addresses an important question in the field. However, my main concern is whether a schema-facilitated motor memory consolidation is possible with the current design as the authors do not control for task interference. The authors state that “Sequence 2 was highly similar to Sequence 1, as 75% of the elements were in the same ordinal position. The positions of keys 2 and 3 (represented in red) were switched in Sequence 2, resulting in 4 novel key transitions to be learned relative to Sequence 1.” Is changing the position of 2 keys enough to consolidate the new sequence as a separate memory trace that is compatible with and integrated into a pre-existing schema? Or would the learning of new sequences result in an updated representation of the original one? I strongly suggest the authors discuss the results in the context of task interference between competing memory traces.

Major comments:

Line 32 & 72: the authors investigate the relationship between NREM sleep duration and post-sleep behavioural performance. However, the results of reference (6) suggest that REM sleep “plays a crucial role in the rapid consolidation of schema-conformant items”. Thus, it is unclear why the relationship with REM sleep was not investigated and I suggest the authors test these correlations as well.

Line 161-164: sleep is known to benefit implicitly learned SRTT but only if the learning is contextual (Spencer et al., 2006, Current Biology). The methodological section of the manuscript suggests that the cues were all squares and had the same colour during practice blocks. Have the authors made sure to include contextual association in their task? The lack of such could have contributed to the lack of group effect and should be discussed.

Line 212-216, 442 and 515: please clarify what was the aim of running the generation task. Furthermore, the generation task employed by the authors does not allow to assess the explicit memory of the sequence. Rather, the participants were asked to self-generate the sequence using the same keys that they used to perform the SRTT. Thus, the sequence could have been generated implicitly during the generation task, even in the absence of visual cues. To test the explicit memory of a sequence, one would have to eliminate the sequence-related motor system activity, e.g., by asking the participants to mark sequence order on printed out screenshots of the task, as in Cousins et al. (2014). Thus, I would suggest referring to the task as ‘generation task’ and avoid using terms like ‘explicit awareness’ (line 515 and 441).

Minor comments:

Please state the sample size for each analysis in the figure/table legend.

Line 71: it would be helpful to know what the authors mean by ‘enhanced integration’ here – is it faster/more accurate behavioural performance on the second sequence? If so, I suggest clarifying this in brackets.

Line 102-104: “The primary effect of interest in this earlier work was the group difference in performance on the novel transitions learned in the Session 2 training phase, reflecting integration processes.” - consider rephrasing this sentence as it is hard to grasp.

Line 87-120: please be consistent with the use of words vs numerals for numbers.

Line 132: if I’m not mistaken, this is the first time when the term “SRTT” is introduced. If so, please explain the abbreviation here and use “SRTT” (instead of “SRT task”, e.g., line 372, 380) throughout the manuscript.

Line 136: replace “design: participants” with “design. Participants” for consistency with line 139.

Line 143: to avoid confusion, please specify in the figure legend that the thumbs were excluded from the numbering system.

Line 234: why were channels Fz, PZ and Oz omitted during habituation?

Line 235: what do the authors mean by “a supplemental individual EEG channel”?

Line 384-385: it is a bit confusing to introduce new exclusion criteria in the figure legend. I suggest moving that sentence to the relevant paragraph in the methods section and then just stating the final sample size for each session in the figure legend.

Line 389: consider referring to Table 3A in the brackets as well “(Figure 2, Session 1; Table 3A)”.

Line 411: it is perhaps worth noting here that the authors also found a significant block effect for accuracy on novel but not learned transitions, perhaps pointing to a decreasing difficulty of the novel transitions over time.

Line 424: the authors might want to make the order of the results consistent with that of the results reported above, i.e., F(1,48)=2.58, p=0.12, n2=0.051 (currently the order of p and n2 is flipped).

Line 428: it is a bit unclear which results this paragraph refers to. Please refer to Table 3 in brackets to avoid such confusion.

Table S8, table legend: the authors state that “No significant Group, Session or Group by Session effects were revealed”, whereas in fact the effect of session was revealed for PVT. The authors note this in the next sentence but I believe the former one should be edited to reflect this as well. Furthermore, the authors say “Note that the significant effect of Session for the measure PVT was driven by overall decreased response time in Session 2 (276 ± 44 ms, compared with 286 ± 55 ms in Session 1), likely reflecting increased familiarity with the task.” – I am not convinced that a decrease in PVT can reflect increased familiarity with the task. If that was indeed the case, then the PVT results should follow the same pattern in Experiment 1, which they do not. Consider rephrasing or adding appropriate references which would back up such claims.

Line 470: please refer to Table 4A in the brackets as well.

Line 474: the authors state that “Accuracy remained stable and high during both training and test runs, with no differences across groups”. However, according to Table 4, block had a significant effect on accuracy at Session 1 Test. Please acknowledge that in text.

Line 497: consider replacing ‘Table 4’ with ‘Table 4B-C’

Line 551-553: As stated by the authors, previous studies have illustrated the involvement of hippocampus in motor sequence learning. However, the SRTT is not purely procedural and includes a declarative component as well (see Robertson 2007, J Neurosci). Thus, I am unsure whether stating that “neuroimaging studies have illustrated the involvement of the hippocampal formation in motor sequence learning” supports the argument that “consolidation processes of declarative and motor memories share common features and rely on similar neural substrates and networks” – the SRTT is just a combination of both, isn’t it?

Line 525: please replace ‘in motor performance’ with ‘in post-sleep motor performance’ or similar.

Reviewer #2: This is a very rigorous and well-conducted experiment. The only issue I noticed is that the link to the raw data currently requires permission to access, so it is not publicly available.

Reviewer #3: In this study Reverberi and colleagues tested in two experiments whether diurnal or nocturnal sleep after the acquisition of a motor schema would facilitate the integration of new, schema-congruent motor information. Interestingly, (when compared to wakefulness) sleep did not exert any beneficial effect on the integration of novel motor information. In addition, there was no significant relationship between NREM sleep features and memory performance.

This is an interesting and timely study. The manuscript is well written and the applied methods are sound. I just have some minor points to add:

1.) The authors state that the computed densities and amplitudes of spindles and slow oscillations were averaged across channels Fz, Cz, and Pz. What is the rationale of this procedure? Why not just using Fz for SOs (were they are dominant) and Cz for spindles?

2.) The authors state that all participants reached sleep stage N2. How many of the participants reached SWS and REM? If the authors would split their sample into participants reaching (for example) SWS and those who did not, would memory performance differ between the groups?

6. PLOS authors have the option to publish the peer review history of their article (what does this mean?). If published, this will include your full peer review and any attached files.

Reviewer #1: No

Reviewer #2: No

Reviewer #3: No

---

## [Author Response · Author response to Decision Letter 0]

23 Nov 2022

Please see appended "Response to Reviewers" document.

---

## [Editor Report · Decision Letter 1]

4 Jan 2023

Sleep does not influence schema-facilitated motor memory consolidation

PONE-D-22-16102R1

Dear Dr. ALBOUY,

In my judgement you have made an excellent job on addressing comments on the manuscript. I am pleased to inform you that your manuscript has been judged scientifically suitable for publication and will be formally accepted for publication once it meets all outstanding technical requirements.

Kind regards,

Shane Lindsay

Academic Editor

PLOS ONE

---

## [Editor Report · Acceptance letter]

6 Jan 2023

PONE-D-22-16102R1 

Sleep does not influence schema-facilitated motor memory consolidation 

Dear Dr. ALBOUY:

I'm pleased to inform you that your manuscript has been deemed suitable for publication in PLOS ONE. Congratulations! Your manuscript is now with our production department. 

Kind regards, 

on behalf of

Dr. Shane Lindsay 

Academic Editor

PLOS ONE